# REPRESENTATIONAL CORRELATES OF HIERARCHICAL PHRASE STRUCTURE IN DEEP LANGUAGE MODELS

## ABSTRACT

While contextual representations from pretrained Transformer models have set a new standard for many NLP tasks, there is not yet a complete accounting of their inner workings. In particular, it is not entirely clear what aspects of sentence-level syntax are captured by these representations, nor how (if at all) they are built along the stacked layers of the network. In this paper, we aim to address such questions with a general class of interventional, input perturbation-based analyses of representations from Transformers networks pretrained with self-supervision. Importing from computational and cognitive neuroscience the notion of representational invariance, we perform a series of probes designed to test the sensitivity of Transformer representations to several kinds of structure in sentences. Each probe involves swapping words in a sentence and comparing the representations from perturbed sentences against the original. We experiment with three different perturbations: (1) random permutations of $n$-grams of varying width, to test the scale at which a representation is sensitive to word position; (2) swapping of two spans which do or do not form a syntactic phrase, to test sensitivity to global phrase structure; and (3) swapping of two adjacent words which do or do not break apart a syntactic phrase, to test sensitivity to local phrase structure. We also connect our probe results to the Transformer architecture by relating the attention mechanism to syntactic distance between two words. Results from the three probes collectively suggest that Transformers build sensitivity to larger parts of the sentence along their layers, and that hierarchical phrase structure plays a role in this process. In particular, sensitivity to local phrase structure increases along deeper layers. Based on our analysis of attention, we show that this is at least partly explained by generally larger attention weights between syntactically distant words.[1]

## 1 INTRODUCTION AND RELATED WORK

It is still unknown how distributed information processing systems encode and exploit complex relational structures in data. The fields of deep learning (Saxe et al., 2013; Hewitt & Manning, 2019), neuroscience (Sarafyazd & Jazayeri, 2019; Stachenfeld et al., 2017), and cognitive science (Elman, 1991; Kemp & Tenenbaum, 2008; Tervo et al., 2016) have given great attention to this question, including a productive focus on the potential models and their implementations of hierarchical tasks, such as predictive maps and graphs.

Natural (human) language provides a rich domain for studying how complex hierarchical structures are encoded in information processing systems. More so than other domains, human language is unique in that its underlying hierarchy has been extensively studied and theorized in linguistics, which provides source of "ground truth" structures for stimulus data. Much prior work on characterizing the types of linguistic information encoded in computational models of language such as neural networks has focused on supervised readout probes, which train a classifier on top pretrained models to predict a particular linguistic label (Belinkov & Glass, 2017; Liu et al., 2019a; Tenney et al., 2019). In particular, Hewitt & Manning (2019) apply probes to discover linear subspaces that encode tree-distances as distances in the representational subspace, and Kim et al. (2020) show that these distances can be used even without any labeled information to induce hierarchical structure. However, recent work has highlighted issues with correlating supervised probe performance with the amount

---

[1]Datasets, extracted features and code will be publicly available upon publication.

of language structure encoded in such representations (Hewitt & Liang, 2019). Another popular approach to analyzing deep models is through the lens of geometry (Reif et al., 2019; Gigante et al., 2019). While geometric interpretations provide significant insights, they present another challenge in summarizing the structure in a quantifiable way. More recent techniques such as replica-based mean field manifold analysis method (Chung et al., 2018; Cohen et al., 2019; Mamou et al., 2020) connects representation geometry with linear classification performance, but the method is limited to categorization tasks.

In this work, we make use of an experimental framework from cognitive science and neuroscience to probe for hierarchical structure in contextual representations from pretrained Transformer models (i.e., BERT (Devlin et al., 2018) and its variants). A popular technique in neuroscience involves measuring change in the population activity in response to controlled, input perturbations (Mollica et al., 2020; Ding et al., 2016). We apply this approach to test the characteristic scale and the complexity (Fig. 1) of hierarchical phrase structure encoded deep contextual representations, and present several key findings:

1. Representations are distorted by shuffling small $n$-grams in early layers, while the distortion caused by shuffling large $n$-grams starts to occur in later layers, implying the scale of characteristic word length increases from input to downstream layers.

2. Representational distortion caused by swapping two constituent phrases is smaller than when the control sequences of the same length are swapped, indicating that the BERT representations are sensitive to hierarchical phrase structure.

3. Representational distortion caused by swapping adjacent words across phrasal boundary is larger than when the swap is within a phrasal boundary; furthermore, the amount of distortion increases with the syntactic distance between the swapped words. The correlation between distortion and tree distance increases across the layers, suggesting that the characteristic complexity of phrasal subtrees increases across the layers.

4. Early layers pay more attention between syntactically closer adjacent pairs and deeper layers pay more attention between syntactically distant adjacent pairs. The attention paid in each layer can explain some of the emergent sensitivity to phrasal structure across layers.

Our work demonstrates that interventional tools such as controlled input perturbations can be useful for analyzing deep networks, adding to the growing, interdisciplinary body of work which profitably adapt experimental techniques from cognitive neuroscience and psycholinguistics to analyze computational models of language (Futrell et al., 2018; Wilcox et al., 2019; Futrell et al., 2019; Ettinger, 2020).

## 2 METHODS

Eliciting changes in behavioral and neural responses through controlled input perturbations is a common experimental technique in cognitive neuroscience and psycholinguistics (Tsao & Livingstone, 2008; Mollica et al., 2020). Inspired by these approaches, we perturb input sentences and measure the discrepancy between the resulting, perturbed representation against the original. While conceptually simple, this approach allows for a targeted analysis of internal representations obtained from different layers of deep models, and can suggest partial mechanisms by which such models are able to encode linguistic structure. We note that sentence perturbations have been primarily utilized in NLP for representation learning (Hill et al., 2016; Artetxe et al., 2018; Lample et al., 2018), data augmentation (Wang et al., 2018; Andreas, 2020), and testing for model robustness (e.g., against adversarial examples) (Jia & Liang, 2017; Belinkov & Bisk, 2018). A methodological contribution of our work is to show that input perturbations can serve as a useful tool for analyzing representations learned by deep networks.

### 2.1 SENTENCE PERTURBATIONS

In this work we consider three different types of sentence perturbations designed to probe for different phenomena.

$n$-**gram shuffling** In the $n$-gram shuffling experiments, we randomly shuffle the words of a sentence in units of $n$-grams, with $n$ varying from 1 (i.e., individual words) to 7 (see Fig. 2a for an example).

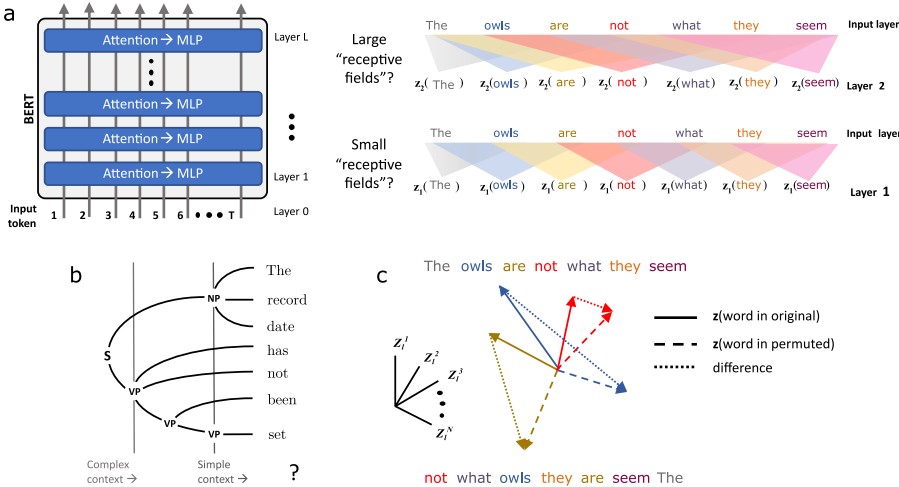

Figure 1: Do Transformers build complexity along their layers? (**a**) The representation of a word is a function of its context, and this cartoon illustrates an hypothesis that deeper representations use larger contexts. (**b**) An example parse tree, illustrating our notion of phrase complexity. (**c**) Cartoon of the distortion metric, where vectors are the z-scored feature vectors **z**, and color map vectors to words.

While the number of words which change absolute position is similar for different $n$, larger $n$ will better preserve the local context (i.e., relative position) of more words. Thus, we reason that $n$-gram swaps affect the representations selective to the context with size $n$ or higher within the sentence, and that lower $n$ will result in greater distortion in sentence representations.

**Phrase swaps** The $n$-gram shuffling experiments probe for sensitivity of representations to local context without taking into account syntactic structure. In the phrase swap experiments, we perturb a sentence by swapping two randomly chosen spans. We explore two ways of swapping spans. In the first setting, the spans are chosen such that they are valid phrases according to its parse tree.[2] In the second setting, the spans are chosen that they are invalid phrases. Importantly, in the second, control setting, we fix the length of the spans such that the lengths of spans that are chosen to be swapped are the same as in the first setting (see Fig. 3a for an example). We hypothesize that swapping invalid phrases will result in more distortion than swapping valid phrases, since invalid swaps will result in greater denigration of syntactic structure.

**Adjacent word swaps** In the adjacent word swapping experiments, we swap two adjacent words in a sentence. We again experiment with two settings – in the first setting, the swapped words stay within the phrase boundary (i.e., the two words share the same parent), while in the second setting, the swapped words cross phrase boundaries. We also perform a more fine-grained analysis where we condition the swaps based on the "syntactic distance" between the swapped words, where syntactic distance is defined as the distance between the two words in the parse tree (see Fig. 4c). Since a phrase corresponds to a subtree of the parse tree, this distance also quantifies the number of nested phrase boundaries between two adjacent words. Here, we expect the amount of distortion to be positively correlated with the syntactic distance of the words that are swapped.

## 2.2 CONTEXTUAL REPRESENTATIONS FROM TRANSFORMERS

For our sentence representation, we focus on the Transformer-family of models pretrained on large-scale language datasets (BERT and its variants). Given an input word embedding matrix $\mathbf{X} \in \mathbb{R}^{T \times d}$ for a sentence of length $T$, the Transformer applies self attention over the previous layer's representation to produce a new representation,

$$\mathbf{X}_l = f_l([\mathbf{H}_{l,1}, \ldots, \mathbf{H}_{l,H}]), \qquad \mathbf{H}_{l,i} = \mathbf{A}_{l,i} \mathbf{X}_{l-1} \mathbf{V}_{l,i},$$
$$\mathbf{A}_{l,i} = \text{softmax}\left(\frac{(\mathbf{X}_{l-1}\mathbf{Q}_{l,i})(\mathbf{X}_{l-1}\mathbf{K}_{l,i})^{\top}}{\sqrt{d_k}}\right), \tag{1}$$

---

[2]We use constituency parse trees from the English Penn Treebank (Marcus et al., 1994).

where $f_l$ is an MLP layer, $H$ is the number of heads, $d_H = \frac{d}{H}$ is the head embedding dimension, and $\mathbf{Q}_{l,i}, \mathbf{K}_{l,i}, \mathbf{V}_{l,i} \in \mathbb{R}^{d \times d_k}$ are respectively the learned query, key, and value projection matrices at layer $l$ for head $i$. The MLP layer consists of a residual layer followed by layer normalization and a nonlinearity. The 0-th layer representation $\mathbf{X}_0$ is obtained by adding the position embeddings and the segment embeddings to the input token embeddings $\mathbf{X}$, and passing it through normalization layer.[3]

In this paper, we conduct our distortion analysis mainly on the intermediate Transformer representations $\mathbf{X}_l = [\mathbf{x}_{l,1}, \ldots, \mathbf{x}_{l,T}]$, where $\mathbf{x}_{l,t} \in \mathbb{R}^d$ is the contextualized representation for word $t$ at layer $l$.[4] We analyze the trend in distortion as a function of layer depth $l$ for the different perturbations. We also explore the different attention heads $\mathbf{H}_{l,i} \in \mathbb{R}^{T \times d_H}$ and the associated attention matrix $\mathbf{A}_{l,i} \in \mathbb{R}^{T \times T}$ to inspect whether certain attention heads specialize at encoding syntactic information.

## 2.3 DISTORTION METRIC

Our input manipulations allow us to specify the distortion at the input level, and we wish to measure the corresponding distortion in the representation space (Fig. 1). Due to the attention mechanism, a single vector in an intermediate layer is a function of the representations of (potentially all) the other tokens in the sentence. Therefore, the information about a particular word might be distributed among the many feature vectors of a sentence, and we wish to consider all feature vectors together as a single sentence-level representation.

We thus represent each sentence as a matrix and use the distance induced by matrix 2-norm. Specifically, let $\mathbf{P} \in \{0,1\}^{T \times T}$ be the binary matrix representation of a permutation that perturbs the input sentence, i.e., $\tilde{\mathbf{X}} = \mathbf{PX}$. Further let $\tilde{\mathbf{X}}_l$ and $\mathbf{X}_l$ be the corresponding sentence representations for the $l$-th layer for the perturbed and original sentences. To ignore uniform shifting and scaling, we also z-score each feature dimension of each layer (by subtracting the mean and dividing by the standard deviation where these statistics are obtained from the full Penn Treebank corpus) to give $\tilde{\mathbf{Z}}_l$ and $\mathbf{Z}_l$. Our distortion metric for layer $l$ is then defined as $\|\mathbf{Z}_l - \mathbf{P}^{-1}\tilde{\mathbf{Z}}_l\|/\sqrt{Td}$, where $\|\cdot\|$ is the matrix 2-norm (i.e., Frobenius norm).[5] Importantly, we invert the permutation of the perturbed representation to preserve the original ordering, which allows us to measure the distortion due to structural change, rather than distortion due to simple differences in surface form. We divide by $\sqrt{Td}$ to make the metric comparable between sentences (with different $T$) and networks (with different $d$).

Intuitively, our metric is the scaled Euclidean distance between the z-scored, flattened sentence representations, $\mathbf{z}_l \in \mathbb{R}^{Td}$. Because each dimension is independently centered and standardized, the maximally unstructured distribution of $\mathbf{z}_l$ is an isotropic $Td$-dimensional Gaussian. The expected distance between two such vectors is approximately $\sqrt{2Td}$. Therefore, we can interpret a distortion value approaching $\sqrt{2}$ as comparable to if we had randomly redrawn the perturbed representation.

## 3 EXPERIMENTAL SETUP

We apply our perturbation-based analysis on sentences from the English Penn Treebank (Marcus et al., 1994), where we average the distortion metric across randomly chosen sentences (see Sec. A.1 for the exact details). We analyze the distortion, as measured by length-normalized Frobenius norm between the perturbed and original representations, as a function of layer depth. Layers that experience large distortion when the syntactic structure is disrupted from the perturbation can be interpreted as being more sensitive to hierarchical syntactic structure.

As we found the trend to be largely similar across the different models, in the following section, we primarily discuss results from BERT (`bert-base-cased`), which has 12 layers and hidden size of 768 (Devlin et al., 2018). We show results from other pretrained and randomly-initialized Transformer-based models, including RoBERTa (Liu et al., 2019b), ALBERT (Lan et al., 2019), DistilBERT (Sanh et al., 2019), and XLNet (Yang et al., 2019), in the appendix (Sec. A.2).

---

[3]However, the exact specification for the MLP and $\mathbf{X}_0$ may vary across different pretrained models.

[4]BERT uses BPE tokenization (Sennrich et al., 2015), which means that some words are split into multiple tokens. Since we wish to evaluate representations at word-level, if a word is split into multiple tokens, its word representation is computed as the average of all its token representations.

[5]There are many possible ways of measuring distortion, such as the average cosine similarity or distance between corresponding feature vectors, as well as different matrix norms. We observed the results to be qualitatively similar for different measures, and hence we focus on the Frobenius norm in our main results. We show the results from additional distortion metrics in Sec. A.3.

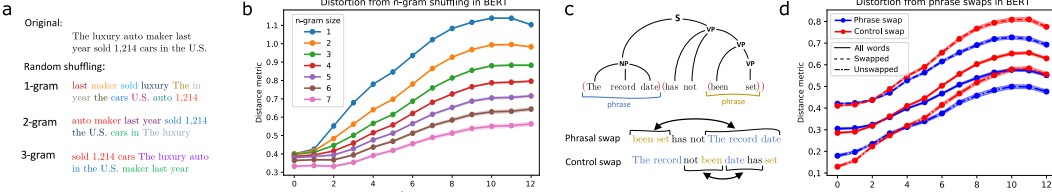

Figure 2: Swapping $n$-grams and phrases. (**a**) Examples of basic $n$-gram shuffles, where colors indicate the units of shuffling. (**b**) Distortion metric computed at each layer, conditioned on $n$-gram size. Error bars hereafter represent standard error across 400 examples. (**c**) An example parse tree, with phrase boundaries shown as grey brackets, and two low-order phrases marked; and examples of a phrasal and control swap, with colors corresponding to the phrases marked above. (**d**) Distortion, computed at each layer, using either the full sentence, the subsentence of unswapped words, or the subsentence of swapped words, conditioned on swap type.

# 4 RESULTS

We summarize our findings for the different perturbations below. While not shown in the main results, we note that randomly-initialized (i.e. untrained) models (somewhat unsuprisingly) exhibit a flat distortion trend for all perturbations (see Sec. A.2). This indicates that the patterns observed here are due to the model's structural knowledge acquired through training, and not simply due to the underlying architecture.

## 4.1 CHARACTERISTIC SCALE INCREASES ALONG BERT LAYERS

When we shuffle in units of larger $n$-grams, it only introduces distortions in the deeper BERT layers compared to smaller $n$-gram shuffles. The $n$-gram sized shuffles break contexts larger than $n$, while preserving contexts of size $n$ or smaller. Interestingly, smaller $n$-gram shuffles diverge from the original sentence in the early layers (Fig. 2b, top curve), implying that only in early layers are representations built from short-range contexts. Larger $n$-gram shuffles remain minimally distorted for 'longer' (Fig. 2b, bottom curve), implying that long-range contexts play a larger role deeper layer representations.

**Phrasal boundaries matter**    Since BERT seems to build larger contexts along its layers, we now ask whether those contexts are structures of some grammatical significance. A basic and important syntactic feature is the constituent phrase, which BERT has previously been shown to represented in some fashion (Goldberg, 2019; Kim et al., 2020). We applied two targeted probes of phrase structure in the BERT representation, and found that phrasal boundaries are indeed influential.

If we swap just two $n$-grams, the BERT representations are less affected when phrases are kept intact. We show this by swapping only two $n$-grams per sentence and comparing the distortion when those $n$-grams are phrases to when they cross phrase boundaries (Fig. 3a), where we control for the length of $n$-grams that are swapped in both settings. There is less distortion when respecting phrase boundaries. Furthermore, the distortion is evident among all feature vectors, including those in the position of words which did not get swapped (Fig. 2d). The global contextual information, distributed across the sentence, is affected by the phrase boundary.

## 4.2 PHRASE HIERARCHY MATTERS

Having seen that representations are sensitive to phrase boundaries, we next explore whether that sensitivity is proportional to the number of phrase boundaries that are broken, which is a quantity related to the phrase hierarchy. Instead of swapping entire phrases, we swap two adjacent words and analyze the distortion based on how far apart the two words are in the constituency tree (Fig. 3a)[6]. This analysis varies the distance in the deeper tree structure while keeping the distance in surface form constant (since we always swap adjacent words).

If the hierarchical representations are indeed being gradually built up along the layers of these pretrained models, we expect distortion to be greater for word swaps that are further apart in tree distance. We indeed find that there is a larger distortion when swapping syntactically distant words (Fig. 3b). This distortion grows from earlier to later BERT layers. Furthermore, when looking at the per-head representations of each layer, we see that in deeper layers there are more heads showing a

---

[6]Note that for adjacent words, the number of broken phrase boundaries equals the tree distance minus two.

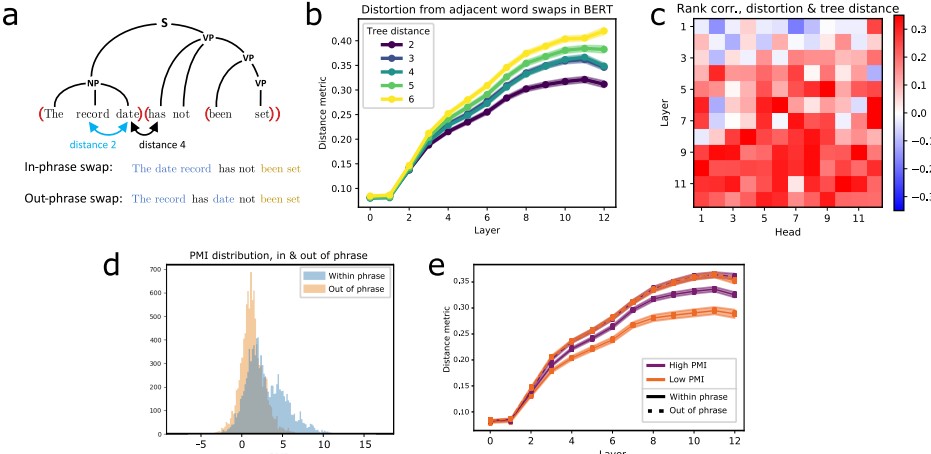

Figure 3: Syntactic distance affects representational distortion. (**a**) An example of adjacent swaps which do and do not cross a phrase boundary, with low-order phrases colored. Phrase boundaries are drawn in red. (**b**) Distortion in each layer, but conditioned on the tree distance. (**c**) For each head (column) of each layer (row), the (Spearman) rank correlation between distortion and tree distance of the swapped words. Colors are such that red is positive, blue negative. (**d**) Histogram of PMI values, for pairs in the same phrase and not. (**e**) Similar to **b**, but averaging all out-of-phrase swaps, and separating pairs above ('high') or below ('low') the median PMI.

positive rank correlation between distortion and tree distance (Fig. 3c). In addition to a sensitivity to phrase boundaries, deeper BERT layers develop a sensitivity to the number of boundaries that are broken.

**Controlling for co-occurrence**   Since words in the same phrase may tend to occur together more often, co-occurrence is a potential confound when assessing the effects of adjacent word swaps. Co-occurrence is a simple statistic which does not require any notion of grammar to compute – indeed it is used to train many non-contextual word embeddings (e.g., word2vec Mikolov et al. (2013), GloVe Pennington et al. (2014)). So it is natural to ask whether BERT's resilience to syntactically closer swaps goes beyond simple co-occurrence statistics. For simplicity, let us focus on whether a swap occurs within a phrase (tree distance = 2) or not.

As an estimate of co-occurrence, we used the pointwise mutual information (PMI). Specifically, for two words $w$ and $v$, the PMI is $\log \frac{p(w,v)}{p(w)p(v)}$, which is estimated from the empirical probabilities. We confirm that adjacent words in the same phrase do indeed have a second mode at high PMI (Fig. 3d). Dividing the swaps into those whose words have high PMI (above the marginal median) and low PMI (below it), we can see visually that the difference between within-phrase swaps and out-of-phrase swaps persists in both groups (Fig. 3e). For a more careful statistical test, in the appendix we show results from running a linear regression between distortion and the phrase boundary which accounts for dependency on any smooth function of PMI (see details in A.4). Even when accounting for the effect of PMI, there is a significant correlation between the breaking of a phrase and the subsequent distortion. This indicates that the greater distortion for word swaps which cross phrase boundaries is not simply due to surface co-occurrence statistics.

**Effects on linguistic information**   Do our input perturbations, and the resulting the distortions, reflect changes in the encoding of important linguistic information? One way to address this question, which is popular in computational neuroscience (DiCarlo & Cox, 2007) and more recently BERTology (Liu et al., 2019a; Tenney et al., 2019), is to see how well a linear classifier trained on a linguistic task generalizes from the (representations of the) unperturbed sentences to the perturbed ones. With supervised probes, we can see how much the representations change with respect to the subspaces that encode specific linguistic information.

Specifically, we relate representational distortion to three common linguistic tasks of increasing complexity: part of speech (POS) classification; grandparent tag (GP) classification (Tenney et al.,

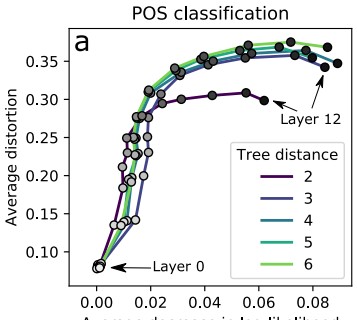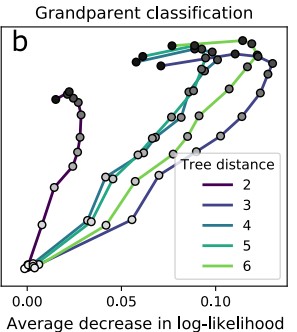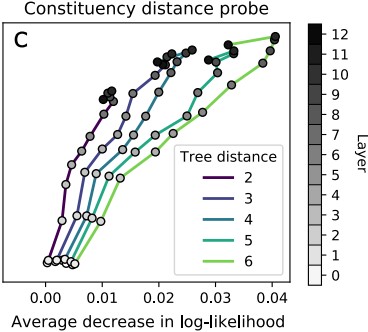

Figure 4: Distortion and inference impairment for increasing linguistic complexity. In each plot, a point is the average (distortion, 'impairment') for a given layer and a given class of word swap distance. Points are connected by lines according to their swap type (i.e. tree distance). The circles are colored according to layer (see right for a legend). Averages are taken over 600 test sentences, with one of each swap type per sentence, and both distortion and log-likelihood are computed for every word in the sentence.

2019); and a parse tree distance reconstruction (Hewitt & Manning, 2019)[7]. The probe trained on each of these tasks is a generalized linear model, linearly mapping a datapoint $\mathbf{x}$ (i.e. BERT representations from different layers) to a conditional distribution of the labels, $p(y|\theta(\mathbf{x}))$ (see A.5 for more details). Thus a ready measure of the effect of each type of swap, for a single sentence, is $\log p(y|\theta(\mathbf{x}_i)) - \log p(y|\theta(\tilde{\mathbf{x}}_i))$, where $\tilde{\mathbf{x}}_i$ is same datum as $\mathbf{x}_i$ in the perturbed representation[8]. Averaging this quantity over all datapoints gives a measure of content-specific distortion within a representation, which we will call "inference impairment".

Based on the three linguistic tasks, the distortion we measure from the adjacent word swaps is more strongly related to more complex information. The inverted L shape of Fig. 4a suggests that increasing distortion is only weakly related to impairment of POS inference, which is perhaps unsurprising given that POS tags can be readily predicted from local context. A deeper syntactic probe, the GP classifier (4b), does show a consistent positive relationship, but only for swaps which break a phrase boundary (i.e. distance >2). Meanwhile, impairment of the distance probe (4c), which reconstructs the full parse tree, has a consistently positive relationship with distortion, whose slope is proportionate to the tree distance of the swap. Thus, when specifically perturbing the phrasal boundary, the representational distortion is related to relatively more complex linguistic information.

### 4.3 A POSSIBLE MECHANISTIC EXPLANATION THROUGH ATTENTION

In the transformer architecture, contexts are built with the attention mechanism. Recall that attention is a mechanism for allowing input vectors to interact when forming the output, and the ultimate output for a given token is a convex combination of the features of all tokens (Eq. 1). While any interactions between inputs must be mediated by attention, it is not obvious how the contextual information of a particular layer is captured by attention in that layer. It has been shown qualitatively that, within a layer, BERT allocates attention preferentially to words in the same phrase (Kim et al., 2020). Our next suite of experiments asks whether this might explain the observed relationship between tree distance and distortion.

We find that in many Transformer heads, the attention—much like distortion—is proportional to the syntactic distance between two words. Fig. 5c summarizes this trend by showing the Spearman rank correlation between the parse tree distance of adjacent word pairs, and the attention paid between those words. Different attention heads within a layer range from correlated to anti-correlated, and with slightly more positively correlated heads in deeper layers. However, there is great variability in this, suggesting that only certain heads learn to specialize to syntactic phenomena.

---

[7]While the original paper predicted dependency tree distance, in this paper we instead predict the constituency tree distance.

[8]POS- and GP-tag prediction outputs a sequence of labels for each sentence, while the distance probe outputs the constituency tree distance between each pair of words. Then $\log p(y|\theta(\mathbf{x}_i))$ is simply the log probability of an individual label.

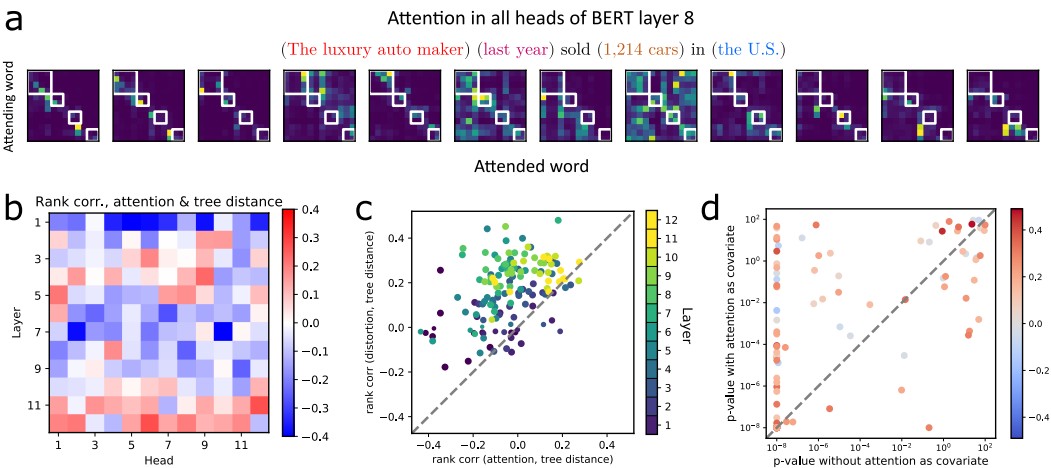

Figure 5: Attention provides a possible explanation for the trends we observe in distortion. (**a**) An example of the attention matrices for all heads in a single layer (layer 8), given the above sentence as input. Phrases in the sentence are drawn as blocks in the matrix. (**b**) The rank correlation of attention vs. tree distance for all the heads/layers. (**c**) The rank correlation coefficients of distortion (y-axis) and attention (x-axis) against tree distance, colored by layer. Marker size is proportional to 1 minus the $p$-value of the distortion/distance correlation. (**d**) A comparison of p-values on the distortion vs. tree distance correlation without (x-axis) and with attention splines as covariates (y-axis).

We observe that at least some of the observed correlation between swap-induced distortion and parse distance can be accounted for by attention. Of course, all interactions between inputs are mediated by attention, but it is not certain that the contextual information in a particular layer comes from the attention at that layer. To test a whether the correlation between tree distance and distortion persists when accounting for attention, we used a linear regression with any smooth function of attention as a covariate (see A.4). We observe larger $p$-values in the controlled regression, indicating that the correlations become less significant when accounting for attention (Fig. 5d). This suggests that the attention in each layer helps to build sensitivity to syntactic distance.

## 5 Discussion

In this paper, we used the representational change in response to perturbed input in order to study the encoding of hierarchical phrasal structure in deep language models. We also identify a link between the perturbation-induced distortion to the magnitude of attention paid to words within and out of phrase boundaries as a potential mechanistic explanation. Across different models, we find that the word-level contexts used to represent a sentence grow in size and complexity along the model layers, similar to the increasing size of receptive fields found in sensory systems.

In neuroscience, it is well accepted that small receptive fields tuned to simple stimuli (i.e., edges) are combined to form larger receptive fields tuned to more complex stimuli (i.e., objects) (Riesenhuber & Poggio, 1999). In language, a phrase within a sentence can serve as a conceptual unit, much like an object in a visual scene, thus motivating our perturbation-based probe for object-like representational sensitivity of phrases. We showed that BERT and its variants are indeed sensitive to the phrasal unit, as demonstrated by greater invariance to perturbations preserving phrasal boundaries compared to control perturbations which break the phrasal boundaries (Fig. 2-5 for BERT, see SM for other models).

Our method and results suggest many interesting future directions. We hope that this work will motivate: (1) a formal theory of efficient hierarchical data representations in distributed features; (2) a search for the causal connection between attention structure, the representational geometry, and the model performance; (3) potential applications in network pruning studies; (4) an extension of the current work as a hypothesis generator in neuroscience to understand how neural populations implement tasks with an underlying compositional structure.

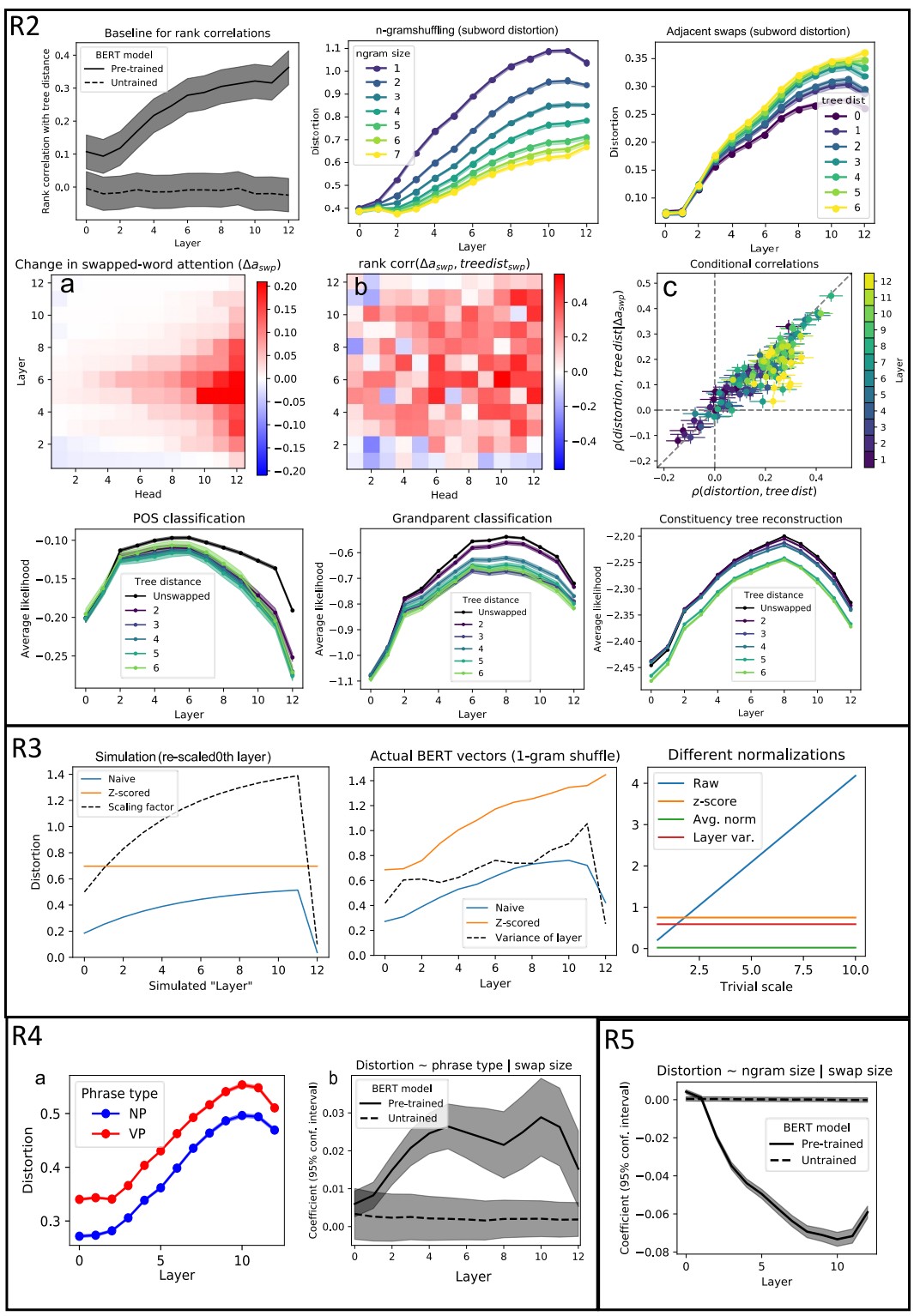

Figure 6: Additional experiments and clarifications (by reviewer).

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

## A    SUPPLEMENTARY MATERIAL (SM)

### A.1    ADDITIONAL DETAILS ON THE DATASET

In this section, we describe additional details of the manipulations done on the datasets.

$n$-**gram shuffling**    For a given a sentence, we split it into sequential non-overlapping $n$-gram's from left to right; if the length of the sentence is not a multiple of $n$, the remaining words form an additional $m$-gram, $m < n$. The list of the $n$-gram's is randomly shuffled. Note that the 1-gram case is equivalent to a random shuffling of the words. In our analysis, we consider $n$-grams, with $n$ varying from 1 (i.e., individual words) to 7 and all the sentences have at least 10 words.

We provide here an example of $n$-gram shuffling.

- Original: The market 's pessimism reflects the gloomy outlook in Detroit
- 1-gram : market pessimism the 's Detroit in The gloomy reflects outlook
- 2-gram : 's pessimism in Detroit The market reflects the gloomy outlook
- 3-gram : The market 's gloomy outlook in pessimism reflects the Detroit

- 4-gram : in Detroit The market 's pessimism reflects the gloomy outlook
- 5-gram : the gloomy outlook in Detroit The market 's pessimism reflects
- 6-gram : outlook in Detroit The market 's pessimism reflects the gloomy
- 7-gram : in Detroit The market 's pessimism reflects the gloomy outlook

**Phrase swaps**  Using constituency trees from the Penn TreebankMarcus et al. (1994), we define phrases as constituents which don't contain any others within them. (See Fig. 2c or Fig. 3a in the main text.) Phrase swaps thus consist of swapping one phrase with another, and leaving other words intact.

To provide an appropriate control perturbation, we swap two disjoint $n$-grams, which are the same size as true phrases but cross phrase boundaries.

**Adjacent word swaps**  To better isolate the effect of broken phrase boundaries, we used adjacent word swaps. Adjacent words were chosen randomly, and one swap was performed per sentence.

## A.2 ADDITIONAL MODELS

### A.2.1 PRE-TRAINED MODELS

We present briefly the pre-trained models that we used for the experiments.[9]

- **BERT** `bert-base-cased`. 12-layer, 768-hidden, 12-heads, 110M parameters.
- **RoBERTa** `roberta-base`. 12-layer, 768-hidden, 12-heads, 125M parameters.
- **ALBERT** `albert-base-v1`. 12 repeating layers, 128 embedding, 768-hidden, 12-heads, 11M parameters.
- **DistilBERT** `distilbert-uncased`. 6-layer, 768-hidden, 12-heads, 66M parameters. The model distilled from the BERT model `bert-base-uncased` checkpoint.
- **XLNet** `xlnet-base-cased`. 12-layer, 768-hidden, 12-heads, 110M parameters.

Note that the hidden size is 768 across all the models. For each pre-trained model, input text is tokenized using its default tokenizer and features are extracted at token level.

### A.2.2 UNTRAINED MODELS

To control for properties which come purely from the architecture, we also compute with randomly-initialized (untrained) models. All model weights are set to a random number. Note that this random initialization has also an impact on the embedding layer.

Here, we provide a side-by-side comparison of results on a trained an untrained model from each model class ($n$-gram: Fig. 7; adjacent: Fig. 8). Across different model classes and tasks, none of our results were replicated in the untrained models. Thus the pattern of invariance we report cannot be explained by architecture alone.

## A.3 ADDITIONAL DISTORTION METRICS

In addition to the scaled Frobenius distance, we considered other ways of measuring distortion in the representation. Here we report results for two different metrics – canonical correlation analysis and a shifted cosine similarity.

**CCA**  Canonical correlations analysis (CCA) Raghu et al. (2017) measures the similarity of two sets of variables using many samples from each. Given two sets of random variables $\mathbf{x} = (x_1, x_2, ..., x_n)$ and $\mathbf{y} = (y_1, y_2, ..., y_m)$, CCA finds linear weights $\mathbf{a} \in \mathbb{R}^n$ and $\mathbf{b} \in \mathbb{R}^m$ which maximise $\mathrm{cov}(\mathbf{a} \cdot \mathbf{x}, \mathbf{b} \cdot \mathbf{y})$. In our context, we treat the representation of the original sentence as $\mathbf{x}$, and the representation of the perturbed sentence as $\mathbf{y}$, and the resulting correlation as a similarity measure.

---

[9]We use the implementation from `https://github.com/huggingface/transformers`.

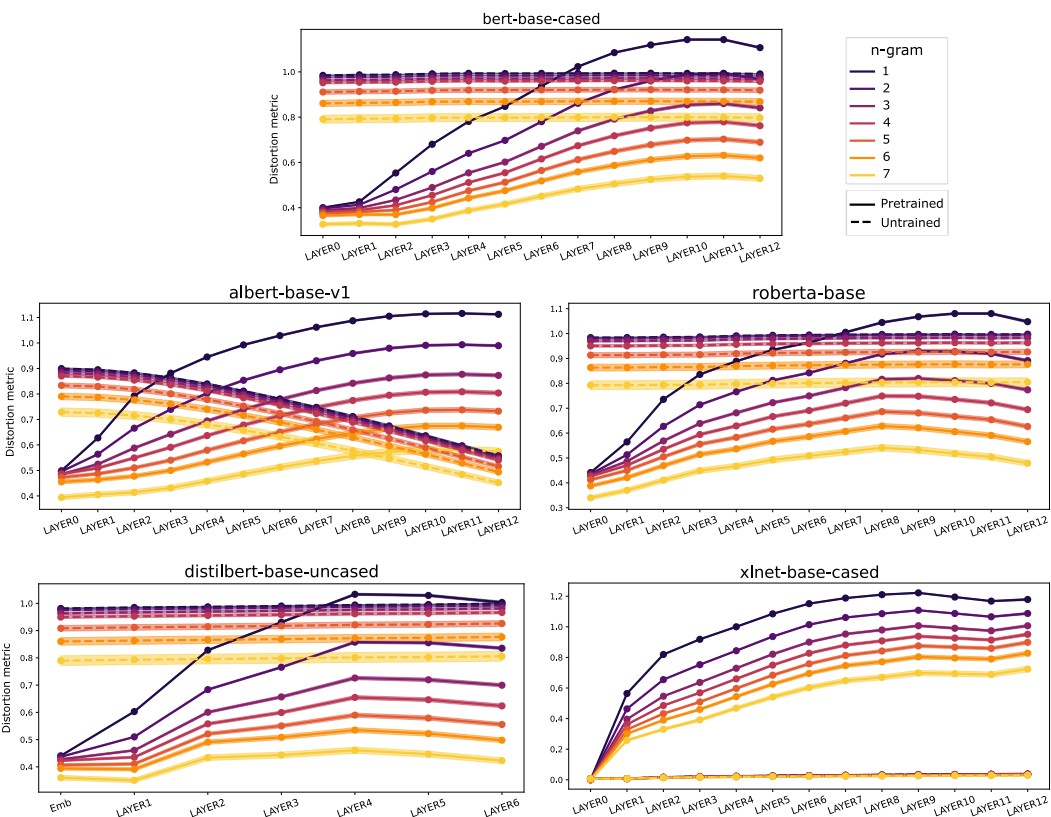

Figure 7: Distortion per layer in $n$-gram shuffling experiments, using different transformer architectures. Lines are the mean Frobenius distance, and the shading is $\pm 1$ standard error of the mean.

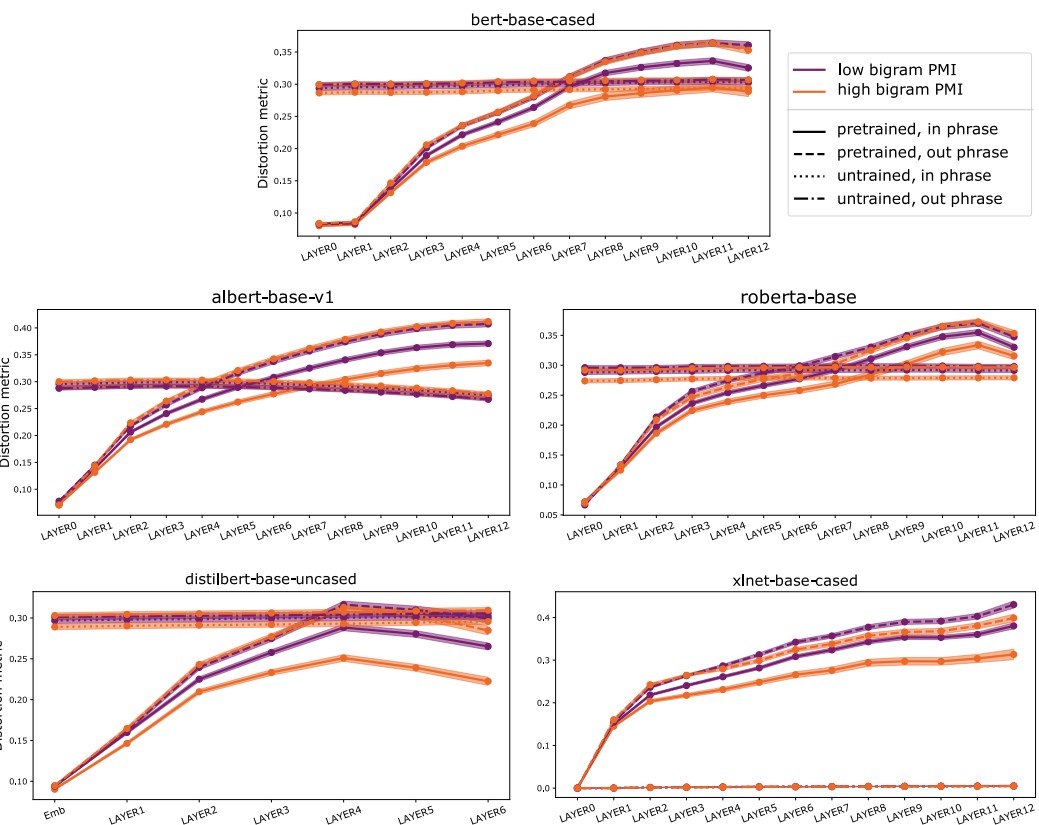

Figure 8: Distortion per layer in the adjacent word swapping experiments, using different transformer architectures. Lines are the mean Frobenius distance, and the shading is ±1 standard error of the mean.

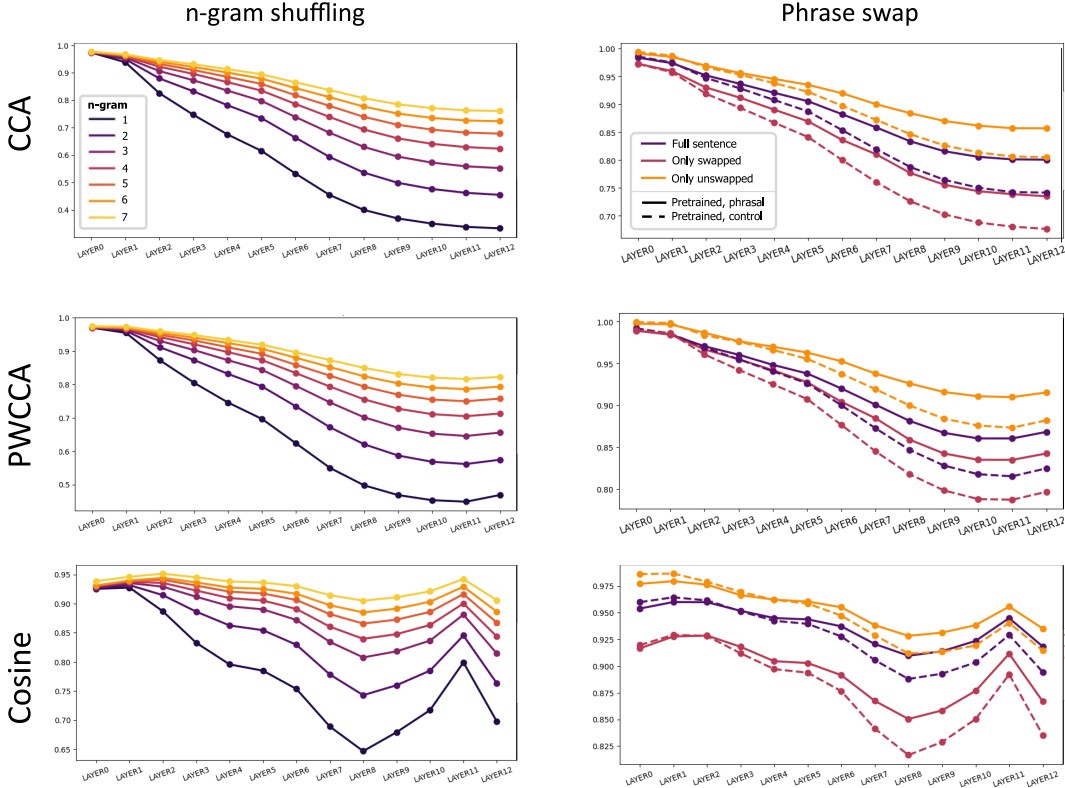

Figure 9: Results from the pretrained BERT model on the $n$-gram shuffling and phrase swap experiments, using alternative distortion metrics described in the text.

Since CCA requires many samples, we use the set of all word-level representations across all perturbed sentences. For example, to construct the samples of $\mathbf{x}$ from $S$ perturbed sentences, we get use $[\mathbf{X}_1|\mathbf{X}_2|...|\mathbf{X}_S]$, where each $\mathbf{X}_i \in \mathbb{R}^{768 \times T_i}$. Unless specified otherwise, $S = 400$. For good estimates, CCA requires many samples (on the order of at least the number of dimensions), and we facilitate this by first reducing the dimension of the matrices using PCA. Using 400 components preserves $\sim 90\%$ of the variance. Thus, while CCA gives a good principled measure of representational similarity, its hunger for samples makes it unsuitable as a per-sentence metric.

We also measured distortion using Projection Weighted Canonical Correlation Analysis (PWCCA), an improved version of CCA to estimate the true correlation between tensors Morcos et al. (2018).[10]

As reported in Figure 9, we did not find any qualitative differences between PWCCA and CCA in our experiments.

**Cosine**  A similarity measure defined on individual sentences is the cosine between the sentence-level representations. By sentence-level representation, we mean the concatenation of the word-level vectors into a single vector $\mathbf{s} \in \mathbb{R}^{NT}$ (where $N$ is the dimension of each feature vector). Treating each dimension of the vector as a sample, we can then define the following metric: $\text{corr}\left(\mathbf{s}_i^{original}, \mathbf{s}_i^{swapped}\right)$. This is equivalent to computing the cosine of the vectors after subtracting the (scalar) mean across dimensions, hence we will refer to it as 'cosine'.

## A.4 PARTIAL LINEAR REGRESSION

In order to control for uninteresting explanations of our results, we often make use of a simple method for regressing out confounds. Generally, we want to assess the linear relationship between $X$ and $Y$,

---

[10]For both CCA and PWCCA, we use the implementation from `https://github.com/google/svcca`.

| | Without PMI | | With PMI | |
|---|---|---|---|---|
| Layer | Coeff.$\times 10^{-2}$ | p-value | Coeff. $\times 10^{-2}$ | p-value |
| Emb. | $-0.21$ | $5.6 \times 10^{-5}$ | $-0.11$ | $9.4 \times 10^{-2}$ |
| 1 | $-0.11$ | $3.4 \times 10^{-2}$ | $-0.05$ | $4.2 \times 10^{-1}$ |
| 2 | $-0.74$ | $< 10^{-16}$ | $-0.53$ | $2.12 \times 10^{-8}$ |
| 3 | $-1.6$ | $< 10^{-16}$ | $-1.3$ | $2.2 \times 10^{-16}$ |
| 4 | $-2.0$ | $< 10^{-16}$ | $-1.4$ | $4.4 \times 10^{-16}$ |
| 5 | $-2.1$ | $< 10^{-16}$ | $-1.5$ | $8.8 \times 10^{-16}$ |
| 6 | $-2.4$ | $< 10^{-16}$ | $-1.7$ | $< 10^{-16}$ |
| 7 | $-2.6$ | $< 10^{-16}$ | $-1.7$ | $1.6 \times 10^{-15}$ |
| 8 | $-3.4$ | $< 10^{-16}$ | $-2.3$ | $< 10^{-16}$ |
| 9 | $-3.8$ | $< 10^{-16}$ | $-2.7$ | $< 10^{-16}$ |
| 10 | $-4.1$ | $< 10^{-16}$ | $-3.0$ | $< 10^{-16}$ |
| 11 | $-3.8$ | $< 10^{-16}$ | $-2.8$ | $< 10^{-16}$ |
| 12 | $-4.2$ | $< 10^{-16}$ | $-3.1$ | $< 10^{-16}$ |

Table 1: Coefficients and p-values of the regular ('without PMI') and controlled ('with PMI') regressions of distortion against phrase boundary.

when accounting for the (potentially non-linear) effect of another variable $Z$. In our experiments, $X$ is always the swap-induced distortion and $Y$ is the swap type, like integer-valued tree distance or binary-valued in/out phrase. We wish to allow $\mathbb{E}[Y|Z]$ and $\mathbb{E}[X|Z]$ to be any smooth function of $Z$, which is achieved by the least-squares solution to the following partially linear model:

$$Y \sim \beta_x X + \beta_{\mathbf{z}} \cdot \mathbf{f}(Z) \iff (Y - \mathbb{E}[Y|Z]) \sim \beta_x(X - \mathbb{E}[X|Z])$$

where $\mathbf{f}(z)$ is a vector of several (we use 10) basis functions (we used cubic splines with knots at 10 quantiles) of $Z$. Both regressions have the same optimal $\beta_x$, but the one on the left is computationally simpler (Hansen, 2000). The standard confidence intervals on $\beta_x$ apply.

Intuitively, the $\beta_x$ obtained by the partially linear regression above is related to the conditional correlation of $X$ and $Y$ given $Z$: $\rho(X, Y|Z)$. Like an unconditonal correlation, it will be zero if $X$ and $Y$ are conditionally independent given $Z$, but not necessarily *vice versa* (both $X$ and $Y$ must be Gaussian for the other direction to be true). To compute conditional rank correlations (which assess a monotonic relationship between $X$ and $Y$), we rank-transform $X$ and $Y$ (this changes the confidence interval calculations).

We apply this method to attentions in Fig. 5. In these supplemental materials, we will also report the results when $X$ is the binary in/out phrase variable, and $Z$ is PMI. The full $p$-values and coefficients of the uncontrolled and controlled regressions can be found in Table 1, where we observe that past layer 2, the $p$-value on phrase boundary is very significant ($p < 10^{-12}$).

## A.5 SUPERVISED PROBES

In this section, we describe the experiments based on the three linguistic tasks: parts of Speech (POS); grandparent tags (GP); and constituency tree distance.

The POS and GP classifiers were multinomial logistic regressions trained to classify each word's POS tag (e.g. 'NNP', 'VB') and the tag of its grandparent in the constituency tree, respectively. If a word has no grandparent, its label is the root token 'S'. The probes were optimized with standard stochastic gradient descent, 50 sentences from the PTB per mini-batch. 10 epochs, at $10^{-3}$ learning rate, were sufficient to reach convergence.

The distance probe is a linear map $\mathbf{B}$ applied to each word-vector $\mathbf{w}$ in the sentence, and trained such that, for all word pairs $i, j$, $\mathrm{TreeDist}(i, j)$ matches $\|\mathbf{B}(\mathbf{w}_i - \mathbf{w}_j)\|_2^2$ as closely as possible. Unlike the classifiers, there is freedom in the output dimension of $\mathbf{B}$; we used 100, although performance and results are empirically the same for any choice greater than $\sim 64$. Our probes are different from Hewitt & Manning (2019) in two ways: (1) we use constituency trees, instead of dependency trees, and (2) instead of an L1 loss function, we use the Poisson (negative) log-likelihood as the loss

function. That is, if $\lambda_{i,j} = \|\mathbf{B}(\mathbf{w}_i - \mathbf{w}_j)\|_2^2$, and $y_{i,j} = \text{TreeDist}(i, j)$

$$-l_{i,j} = y_{i,j} \log \lambda_{i,j} - \lambda_{i,j} - \log y_{i,j}!$$

Otherwise, the probes are trained exactly as in Hewitt & Manning (2019). Specifically, we used standard SGD with 20 sentences from the PTB in each mini-batch, for 40 epochs.

**Evaluation** A linear model is fit to maximize $p(y|\theta(\mathbf{x}))$, with $p$ a probability function (multinomial for classifiers, Poisson for distance), and $\mathbf{x}$ coming from the unperturbed transformer representation. We evaluate the model on $\tilde{\mathbf{x}}$, which are the representations of the data when generated from a perturbed sentence. We take the average of $\log p(y|\theta(\mathbf{x}_i)) - \log p(y|\theta(\tilde{\mathbf{x}}_i))$ over all the data $i$ in all sentences. For example, all words for the classifiers, and all pairs of words for the distance probe. Concretely, we are just measuring the difference in validation loss of the same probe on the $\mathbf{x}$ data and the $\tilde{\mathbf{x}}$ data. But because the loss is an appropriate probability function, we can interpret the same quantity as a difference in log-likelihood between the distribution conditioned on the regular representation and that conditioned on the perturbed representation. Distortion is similarly computed using the full sentence, providing a number for each swap in each sentence.

