# OpenReview forum: "Representational correlates of hierarchical phrase structure in deep language models"
_ICLR.cc/2021/Conference — Reject_

### Official Review · AnonReviewer4 · 2020-10-27
**Reveal the correlations of hierarchical phrase structure and BERT's output, but lack theoretical explanation.**

**Rating:** 6
**Confidence:** 3

**Review:**

--- Overall ---

This paper provides some insights into the relation of BERT's output w.r.t. the parser tree (in terms of constituent phrases) of the input sentence. As some previous work has pointed out, BERT model contains the parsing information (e.g., Hewitt & Manning NAACL’19)). This work can be regarded as a moderate reification and improvement of that thought (but it is still limited to existing scopes and methodologies).

The merits of this paper: (1) the paper reveals some interesting facts such as BERT are sensitive to phrasal hierarchy and there are behavioural discrepancies between different layer; (2) the experiments are comprehensive, including both the distortion analysis and conventional probe approaches.

In my point of view, the main issue of this paper is, like many other works, there is no strong theoretical explanation for the phenomenon being investigated. In this sense, the novelty of this paper is not so strong.

--- Major comments ---

1.	In experiments, it is not clear whether the randomness of BERT itself has been deducted. The randomness could be caused by the dropout operation which may lead to the discrepancy on output even using the same sentence.
2.	In the future version, I recommend to further provide two tests: (1) in current settings, the distortions are showed in sentence-level (e.g., by summing up all distortions within a sentence?). I would like to see a finer-grained test, i.e., whether the most distortion parts are produced by the swapping part; (2) The genre of the Penn treebank dataset is limited to general texts or articles that may be seen in the training corpus of BERT. I would recommend testing on other domains (e.g., biomedical or academic) that BERT never saw before (or structure-less data that do not present a syntactic structure).
3.	Do X~ and X in section 2.3 use the same mean and deviation?
4.	It seems that Fig. 2(c) takes account into both NP and VP, what if we only constrain the phrases to be only NP? Will the distortion becomes large since swapping subject and object will lead to totally different meanings?
5.	There is a lack of explanation about "all words", "swapped", and "unswapped" settings of Fig. 2(d).
6.	Is there any intuition for the step back on the layers 11 and 12 in Fig. 4(b) and 4(c)?

--- Minor comments ---

1.	Strictly speaking, the terminology “gram” in Fig.2(a) should be called "chunk", as “grams” are usually related with sliding window thus overlap with each other.
2.	“Fig.3(a)” in the paragraph just above Section 4.2 should be “Fig.2(c)”.

---

> ### Author Response · Authors · 2020-11-23
> **Review response**
>
> Thanks for your review! Please find our response below:
> - **“In experiments, it is not clear whether the randomness of BERT itself has been deducted. The randomness could be caused by the dropout operation which may lead to the discrepancy on output even using the same sentence.”** Thanks for pointing this out. When BERT is used for inference (as we do in our experiments  when we obtain the representations), dropout is deterministic, so there is no randomness. But we agree that this is an important detail and we will make sure to make this clearer in the paper.
>
> - **“(1) in current settings, the distortions are showed in sentence-level (e.g., by summing up all distortions within a sentence?). I would like to see a finer-grained test, i.e., whether the most distortion parts are produced by the swapping part;”** Thanks for this suggestion! We actually show this in Figure 2(d), where we do indeed see that distortions of swapped words (“swapped”) are much higher than unswapped words (“unswapped”). We will make the legend/caption much clearer in the next version.
>
> - **“(2) The genre of the Penn treebank dataset is limited to general texts or articles that may be seen in the training corpus of BERT. I would recommend testing on other domains (e.g., biomedical or academic) that BERT never saw before (or structure-less data that do not present a syntactic structure).”** One reason why we (and most other BERTology papers) stick to PTB is that PTB is one of the few resources with well-annotated linguistic phenomena, which allows us perform interventional analysis via direct manipulation of syntactic structures. While one could theoretically use off-the-shelf parsers on other domains, this introduces an additional confound in interpreting the results due to error propagation (since  parsers often have trouble generalizing to out-of-domain data). However, the reviewer’s critique (which, again, is applicable to most existing work on analyzing BERT) is well-founded, and in our future work we plan to explore how such probing analyses can be applied to other domains and languages.
>
> - **“Do X~ and X in section 2.3 use the same mean and deviation?”** Yes.
>
> - **“It seems that Fig. 2(c) takes account into both NP and VP, what if we only constrain the phrases to be only NP? Will the distortion becomes large since swapping subject and object will lead to totally different meanings?”** This is a great question. Based on the reviewer's suggestion, we performed a distortion analysis where we only swap NP with NP and VP with VP. (Fig. 6R4 left figure; https://i.imgur.com/B1ImK1q.png). We find that there is greater distortion when VP phrases are swapped. However, this analysis is confounded by the fact that VPs are generally larger. Therefore we performed a multivariate analysis (the method of section A.4 in the supplementary materials) where we regress the distortion against both the type of phrase that is swapped (NP vs VP) and (non-linear functions of) the average size of the phrase that is swapped (Fig. 6R4; right figure). The resulting number can be interpreted as "the expected difference in means, conditional on the swap size". Here, we still observe a greater distortion effect for VPs, although the effect is predictably smaller than in the uncontrolled case and limited to intermediate layers. We think this makes linguistic sense because verbs are considered to be the most important part of a sentence (e.g. in dependency grammars the sentence is headed by the main verb). (In the *untrained* model we found that the swap size accounts for all of the difference between VP and NP, which confirms that those differences are purely superficial.) We thank the reviewer for suggesting this interesting analysis and will include it in the paper. We also observe that the reviewer’s comment demonstrates that our flexible perturbation-based approach naturally lends itself to such questions, and to finer-grained interrogation of results.
>
> - **“There is a lack of explanation about "all words", "swapped", and "unswapped" settings of Fig. 2(d).”** “Swapped” measures the distortion only for words that are swapped, while “Unswapped” measures the distortion only for words that are not swapped (see answer to question 2). We will make this clearer in the legend/caption.
>
> - **Is there any intuition for the step back on the layers 11 and 12 in Fig. 4(b) and 4(c)?** We think that this reversal is happening because in the final few layers, the BERT representations potentially become less contextualized as they have to predict the output word. This phenomena also occurs for supervised probes where task-specific performance often peaks in the middle layers (e.g. in Figure 3 of https://arxiv.org/pdf/1903.08855.pdf).
>
> - **“Minor comments…”** Thanks for the suggestions! We will incorporate them.

---

### Official Review · AnonReviewer2 · 2020-10-28
**Official Blind Review #2**

**Rating:** 6
**Confidence:** 3

**Review:**

### Summary:
This paper addresses how pre-trained Transformer models build their contextual representations along with the layers by measuring changes in the outputs on a series of probes. Specifically, a probe involves swapping words in a sentence and measuring the distortion in the representations. The series of probes are designed to test how the models respond to different syntactic-related swaps including n-gram size and syntactic phrase boundary and distance. The results mainly focus on BERT (base-cased), but some results of the other variants have the same trend. The author first confirms that the observed distortions are the learned behavior by showing a "flat" distortion trend of untrained models. When subjecting to n-gram swaps, the results show that smaller n-grams have a larger distortion. We can also see that later layers are affected more than earlier layers (for all kinds of swaps). Regarding hierarchical phrase structure, the results show that the phrasal boundaries are important, even accounted for PMI of swapped words. The rank correlations between distortion and the syntactic distance are higher in the deeper layers across all attention heads, but the correlations are somewhat weak (around 0.3). In addition, distortions have a larger impact on a relatively more complex task (parse tree distance reconstruction > POS tagging).  Finally, an analysis of the attention shows that the attention layers contribute to the observed distortion. The main conclusion is that BERT (and its variants) builds its contextual representation by increasingly incorporating more phrasal units along with the layers.

### Recommendation:
Overall, I would like to recommend this paper for the conference. The paper presents a novel approach to study the behavior of Transformer models and the findings are quite interesting. My main concerns are that some of the implications of the results are quite hard to follow, and I am not sure how they might impact future downstream researches.

#### Pros:
- I would like to commend the author for clear writing. The paper was very well written and I enjoyed reading the paper.
- The main goal of the paper is to understand (or at least shade some light on) how BERT works. I think it is equally important as the interpretability of the models (i.e., explain the prediction).
- The perturbation-based probes are easily related to language features than other methods such as attention visualization or a salience-based approach. Although there exists work that uses the same approach, I think this paper presents novels finding on how the representations of the pre-trained models "distort" given different perturbations.
- The paper provides comprehensive experiments including many carefully designed experiments to reveal the key insights.


#### Cons:

- Some of the key findings, such as "long-range contexts play a larger role deeper layer representations" are very hard to follow.
- The differences between the distortions among tree distance and the rank correlations are somewhat weak, or rather we do not have a baseline to compare against. I am not sure how strong finding 3 (in the introduction) is supported by the results. Perhaps, to improve the experiment further, the author could include the results of different model architectures (RNN-based models) to compare and contrast.
- I think the experiment on the attention weights is not helpful. Since attention layers are the main mechanism in the Transformer models, it is not surprising that it plays an important role. In the end, it is still inconclusive how attention captures the contextual information. The author could analyze the changes or distortion of attention weights.

### Questions:
1. It is very important how the author handles subwords. The explanation should not be a footnote and there should be more detail on when the "average" happens (i.e., from input to BERT or only distortion computation).
1. I can see why the author would like to aggregate subwords into words when probing. How much do the averaging embeddings change the distortion? Does it affect the findings? Wouldn't it be more helpful to include a probe regarding subwords?
1. What is the corpus that PMI is computed on?
1. What are the base likelihoods (without the perturbation)? Could we imply that the average decrease in log-likelihood is explained by the base likelihoods?

### Comments:
- A.4 is not quite related to the attention covariate, though the intention is understandable.
- Fig. 5e does not exist. I think you mean Fig. 5d (which is quite hard to interpret).
- I think a more careful analysis of the distortion vectors (Fig 1c) should be done (i.e., why is it desirable that the models should be robust to some perturbations? And what is the trend of the distortions across different types of perturbations?).

---

> ### Author Response · Authors · 2020-11-24
> **Review Response**
>
> Thank you for the very helpful review!
> - **“The differences between the distortions among tree distance and the rank correlations are somewhat weak, or rather we do not have a baseline to compare against...”** Important point. We include a comparison of the rank correlations (on the full BERT features) in the pretrained and untrained BERT models (Fig. 6R2; https://i.imgur.com/5fR3Yoq.png). The confidence intervals around the untrained model perhaps provide a lower baseline. Note that the supervised tree-reconstruction probe achieves around 0.8 correlation, which might provide a kind of upper bound as well.
> - **“I think the experiment on the attention weights is not helpful...”** The reviewer raises an important point, which is that the attention changes when we swap the words. Per their suggestion, we include an analysis of those changes (Fig. 6R2 middle row; https://i.imgur.com/Nn7kFzV.png), as a replacement for the existing figure 5. Note that we are only measuring the attention paid between the two swapped words, not all the attentions. In short, we see that heads in middle layers show larger average increase in the attention paid to the swapped words (a). This change is correlated with the tree distance between the words (b, columns sorted as in [a]), and, in a few deeper heads, it can explain some of the correlation between distortion and tree distance (c).
> - **“It is very important how the author handles subwords. The explanation should not be a footnote and there should be more detail on when the "average" happens (i.e., from input to BERT or only distortion computation) .... Does it affect the findings? Wouldn't it be more helpful to include a probe regarding subwords?”** The average happens only for the distortion computation. We find that the distortions do not change appreciably when we compute without averaging subwords (Fig. 6R2; https://i.imgur.com/ueY31Qk.png). We will include a better explanation of how we average them in the main text.
> - **“What is the corpus that PMI is computed on?”** One Billion Word Language Model Benchmark dataset (http://www.statmt.org/lm-benchmark/)
> - **“What are the base likelihoods (without the perturbation)? Could we imply that the average decrease in log-likelihood is explained by the base likelihoods?”** Here (Fig. 6R2; https://i.imgur.com/Vtlw1MI.png) are the average likelihoods for all representations in each layer. To clarify: probes are trained only on the unperturbed representation, and we plot the decrease in generalization performance when testing in each perturbed representation. There is only one base likelihood per probe, and it cannot alone explain the decrease in generalization performance as a function of distortion.
> -**“A.4 is not quite related to the attention covariate, though the intention is understandable.”** Thank you for pointing this out, the method we use is the same but we only explained it in terms of PMI -- that section in the supplemental is now a more general explanation of the regression we used.
> -**“Fig. 5e does not exist. I think you mean Fig. 5d (which is quite hard to interpret).”** Thank you for identifying the typo! We also think that by using the change in local attention, and plotting the confidence intervals instead of p-values, the plot is simpler to interpret.
> -**“I think a more careful analysis of the distortion vectors (Fig 1c) should be done (i.e., why is it desirable that the models should be robust to some perturbations?).”**
> This is a very important and general question, somewhat beyond the scope of our paper. Our goal is less to make a claim about how desirable robustness is, but to develop a measure of a representation’s “awareness” of some linguistic structure (like phrases). In the extreme, for example, if a set of neurons is only representing the phrase structure, then it should be completely robust (zero distortion) to any perturbation which leaves the phrases unaffected, and have non-zero distortion only to phrase-breaking perturbations. In BERT there are certainly multiple linguistic variables being represented simultaneously, but we empirically find (in the paper) that the distortions can still be related to specific linguistic information.
> As far as desirability goes, perhaps there is an analogy to be made with visual networks. In studies of adversarial robustness in visual nets, the amount of distortions in the network output (instead of hidden representations) with respect to targeted input perturbations has been widely used as a tool to interpret which part of the input matters. For example, it’s been reported that adversarially-robust networks are more sensitive to pixel perturbations around object boundaries, than within those boundaries (https://arxiv.org/pdf/1905.09797.pdf), and that non-robust networks do not have this property. Perhaps, in our experiments, a higher robustness to phrase-preserving perturbations indicates that a hidden layer “wishes” to represent phrases more robustly.

---

### Official Review · AnonReviewer1 · 2020-11-03
**Empirical analysis of BERT through swapping words or phrases, no surprising insights**

**Rating:** 5
**Confidence:** 4

**Review:**

This paper analyzes the ability of BERT model to learn good representations of sentences, through a purely empirical study, there is no other contribution of a new model or a new analysis technique. In the experimental analysis, phrases or words are swapped and the corresponding changes in the sentence representations are analyzed from all the hidden layers. I didn't see any surprising outcomes from the analysis, so not sure how impactful this work is. I would rather see this kind of work in a workshop track for ICLR.

A minor comment: only one dataset is considered in analysis. I would like to see the analysis sentences from the training set as well as many test sets.

---

> ### Author Response · Authors · 2020-11-23
> **Review response**
>
> Thanks for the review!
>
> - **"I didn't see any surprising outcomes from the analysis, so not sure how impactful this work is."** While there is indeed some evidence that BERT-like models are able to encode hierarchical syntactic information in its hidden layers [Liu et al. 2019; Tenney et al. 2019; Hewitt and Manning 2019], the field has yet to reach a consensus on (1) methodologies by which to extract such information (prior work has mostly relied on supervised probes which is known to be problematic [Hewitt and Liang 2019]) and (2) the mechanisms by which such information is encoded. We believe that our approach, which measures the change in distributed representations following targeted interventions to specific input structures, is novel, and quite different from the existing work on supervised probes. Finally, even if the phenomenological results are mainly in line findings from existing work on supervised probes, we believe that arriving at these results via a new method is interesting/important!
>
> - **" I would like to see the analysis sentences from the training set as well as many test sets."** One reason why we (and most other BERTology papers) stick to PTB is that PTB is one of the few resources with well-annotated linguistic phenomena, which allows us perform interventional analysis via direct manipulation of syntactic structures. While one could theoretically use off-the-shelf parsers on other domains, this introduces an additional confound in interpreting the results due to error propagation (since parsers often have trouble generalizing to out-of-domain data).

---

### Official Review · AnonReviewer5 · 2020-11-05
**Overall nice, but (i) the distortion measure is questionable and (ii) not clear what to take out of the results**

**Rating:** 5
**Confidence:** 5

**Review:**

The paper investigates the sensitivity of BERT representations to different kinds of permutations in the input sentence. These transformations include n-gram permutation, span swaps (with or without crossing syntactic phrase boundaries), adjacent token swaps (with different syntactic distance). The authors measure the l2 distance between representations coming from original and perturbed input.

Overall, the results suggest that BERT is sensitive to hierarchical phrase structure.

------------------------------------------------------------------------------------------------------
Strengths

The idea of measuring a network’s response to input transformations is nice and potentially could be used to test different kinds of hypotheses.

------------------------------------------------------------------------------------------------------
Weaknesses

My main concern is the measure used for a distance between representations. Namely, this is the l2 distance, which accounts for distance in neurons. Therefore, it does not tell us to what extent representations encode different things but rather how different are their individual neurons. For example, different phenomena can be either focused in a network (encoded by only a few neurons), or distributed (see, e.g., the paper [1] or a more recent [2]).

Therefore,
1) the phenomena that suffer from perturbations have different impact on the l2 distance because they affect different number of neurons
2) as a consequence of the above, it is not clear what to conclude from the proposed results: they are likely to be not because of the high-level differences in what original/perturbed representations encode, but rather in
how these underlying phenomena affect individual neurons.

(To measure differences in what representations encode you can use, for example, PWCCA (NeurIPS 2019 “Insights on representational similarity in neural networks with canonical correlation”) or other related measures.)

[1] AAAI 2019 “What Is One Grain of Sand in the Desert? Analyzing Individual Neurons in Deep NLP Models”

[2] EMNLP 2020 “Analyzing Individual Neurons in Pre-trained Language Models”

Other concerns.
If we put aside the validity of the measure, it is still not clear what to take out of these results: they rather show that the method passes sanity checks rather than tell us something we didn’t know before. For example, there’s a huge line of works showing that BERT “understands” syntax and phrase composition (looking at representations, geometry, attention heads, etc). Hence the results stated in contributions 2, 3, 4 are not surprising. Contribution 1 is also more of a sanity check: of course, shuffling smaller parts has to cause more distortion than shuffling longer phrases.

------------------------------------------------------------------------------------------------------
Questions

Section 4.1, paragraph 1: when referring to figure 2b, you say “When we shuffle in units of larger n-grams, it only introduces distortions in the deeper BERT layers compared to smaller n-gram shuffles.”
I have trouble seeing this from the figure. For all lines, the distortion goes up almost linearly from layer 2 to layers 9-10. Yes, shuffling larger ngrams causes lower distortion, but this is expected. Am I missing something?

------------------------------------------------------------------------------------------------------
Missing references (in addition to mentioned above)

1) When hypothesizing about heads specializing in syntax (section 4.3), none of the relevant previous work is mentioned. For example,
[3] - for NMT Transformer, showed that the important heads are specialized, and many of them are syntactic.
[4] - repeated the previous syntax-heads evaluation for BERT.
[5] - also mention that some heads track syntax.

2) [6] - looks relevant: it also measures changes in representations across models, layers, for different tokens, etc. Maybe the most relevant to your work is the part showing how different words influence each other (e.g., rare words influence others more than frequent ones; the same analysis for POS). Note also which measure they use - pwcca, which is invariant to linear transformations of representations.

[3] ACL 2019 “Analyzing Multi-Head Self-Attention: Specialized Heads Do the Heavy Lifting, the Rest Can Be Pruned”

[4] “Do Attention Heads in BERT Track Syntactic Dependencies?”

[5] BlackBoxNLP 2019 “What Does BERT Look At? An Analysis of BERT's Attention”

[6] EMNLP 2019 “The Bottom-up Evolution of Representations in the Transformer: A Study with Machine Translation and Language Modeling Objectives”

------------------------------------------------------------------------------------------------------
Presentation

The paper is overall fairly clear, but figures 2d and 3e are not readable. Namely, (even with a maximum zoom) it is very hard to distinguish between solid, dashed and dash-dot lines of the same color.

---

> ### Author Response · Authors · 2020-11-23
> **Review response**
>
> Thanks for your review! Please find our response below:
>
> - **“My main concern is the measure used for a distance between representations…”**
> Thank you for raising this important point! We agree that considering more invariant measures of representation similarity is also important, and we actually show results from PWCCA in Figure 9 of the supplementary materials (similar to Figure 2 in the main text, but measured with PWCCA and other measures). We also performed additional experiments with PWCCA for Figure 3, and provided them here (https://imgur.com/a/y02AGbh). In these experiments, we observe qualitatively similar trends between L2 distance and PWCCA. We also note that (1) we are comparing two representations within the same layer, so PWCCA amounts to average feature-level correlations after (scaled) rotations of both representations, and (2) the correlation measure in PWCCA requires multiple samples, limiting the application of the distortion/distance measure to at least a “set of samples”, while L2 distance allows for the measurement for each sentence, which enables further interesting analyses (e.g. Figure 4) . Furthermore, we find that, in our experiments, PWCCA is empirically the same as the average neuron-level correlation (i.e. CCA without the rotations). However, the reviewer raises an important point and we will consider moving the PWCCA analysis to the main part of the paper. We will also add more discussion around this point in the next iteration of the paper.
>
> - **“[the results] are likely to be not because of the high-level differences in what original/perturbed representations encode, but rather in how these underlying phenomena affect individual neurons.”**
> We were also concerned about whether measures of representational dissimilarity (including PWCCA) actually changes in the encoding of linguistic information, and thus, included experiments with supervised probes to tackle precisely that concern. For example, it is possible that all of the perturbations occur in subspaces which are independent of any information. In figure 4 of the main text, we show that, for three tasks of increasing complexity, the adjacent-swap perturbations do produce changes along task-relevant subspaces, but only in the more complex linguistic tasks. Thus, it is not likely that our results are unrelated to differences in encoded information.
>
> - **“For example, there’s a huge line of works showing that BERT “understands” syntax and phrase composition (looking at representations, geometry, attention heads, etc). Hence the results stated in contributions 2, 3, 4 are not surprising.”**
> While there is indeed some evidence that BERT-like models are able to encode hierarchical syntactic information in its hidden layers (Liu et al. 2019; Tenney et al. 2019; Hewitt and Manning 2019), the field has yet to reach a consensus on (1) methodologies by which to extract such information (prior work has mostly relied on supervised probes which is known to be problematic (Hewitt and Liang 2019)) and (2) the mechanisms by which such information is encoded. We believe that our approach, which measures the change in distributed representations following targeted interventions to specific input structures, is novel, and quite different from the existing work on supervised probes. Finally, even if the phenomenological results are mainly in line findings from existing work on supervised probes, we believe that arriving at these results via a new method is interesting/important!
>
> - **“...of course, shuffling smaller parts has to cause more distortion than shuffling longer phrases"**
> This is a valid point. To investigate whether these distortions can be explained via simple superficial differences in position change, we performed multivariate analysis analysis to take into account positional change caused by the n-gram swap. We plot, for the trained and untrained BERTs, the (linear) effect of chunk size on distortion, after accounting for (non-linear) effects of superficial differences (Fig. 6R5; https://imgur.com/Ms3EotS). In *untrained* BERT (which presumably only shows trivial differences), swap size completely accounts for the effect of n-gram size, while in the *pretrained* BERT, the effect of n-gram size is just as strong.
>
> - **"when referring to figure 2b, you say ..."**  It was poor wording on our part to say that distortions were “only introduced” in deeper layers; our point is more that the tiny differences in distortion at the embedding level (superficial differences) become more dramatic in deeper layers.
>
> - **“Missing references (in addition to mentioned above)”** Thanks for the references! We will add them to the paper.
>
> - **“The paper is overall fairly clear, but figures 2d and 3e are not readable.....”** Thank you for the feedback. We will make the visualization clearer in the next iteration of the paper.

---

### Official Review · AnonReviewer3 · 2020-11-10
**Measuring network activations in response to linguistic probes yields expected results**

**Rating:** 6
**Confidence:** 4

**Review:**

The paper investigates the extent to which pre-trained transformer models successfully capture linguistic structure. The approach taken is to present the model with carefully constructed pairs of linguistic probes and then measure the difference in response to a naturally occurring sentence versus one the has been mutated using one of three different strategies. The first is to permute n-grams of a predetermined order. The next is to swap phrases within a parse tree with the results being contrasted with swapping n-grams than don't correspond to phrase boundaries. Finally, the authors explore swapping words that cross varying amounts of syntactic structure.

The part of the paper that I liked best was the introduction of the distortion metric in section 2.3. This seems like it could be a generally useful means for probing networks both for NLP as well as possible for other domains (e.g., CV). The paper would have benefited from spending more time developing and motivating the metric. The construction looks well thought out, however it would have been good to include at least some ablation experiments to show that z-score normalization and including the perturbation matrix in the formulation is necessary for the types of experiments performed later in the paper.  I imagine the z-score normalization is helpful, but I wonder if the probe would also work without the perturbation matrix. If this is the case, it would allow the metric to use to explore manipulations that go beyond word re-orderings.

The experiments in the paper are interesting in that they support the case that large pre-trained transformer models do capture linguistic structure. However, this is also a complementary weakness in that the paper doesn't add anything particularly surprising given prior work in this area.

The paper would be strengthen by introducing additional probes in order to more deeply understand what is an isn't captured by existing pretrained models. I would have also liked to see some treatment of other pretrained models including near BERT variants such as RoBERTa or Albert as well as other models with more distinct architectures and training objectives (e.g., T5 or a GPT model). Alternatively, the paper could have take the approach of developing the best overall probe to detect what is captured by a model. The latter would involve running experiments with different probes and in the best case on different models to discover which method is the most discriminating.

As a minor nite, the paper attempts to make a connection to neuro-science. This would be better done if there was more of a clear and explicit connection between the techniques explored in the paper and existing neuro-science work beyond just the fact the model is using probes and measuring network activity.

---

> ### Author Response · Authors · 2020-11-23
> **Review response**
>
> Thanks for the review! Please find our responses below.
> - **“The construction looks well thought out, however it would have been good to include at least some ablation experiments to show that z-score normalization and including the perturbation matrix in the formulation is necessary for the types of experiments performed later in the paper. I imagine the z-score normalization is helpful, but I wonder if the probe would also work without the perturbation matrix.”** *Permutation matrix.* The permutation matrix is crucial in our framework as ensures that the distortion resulting from the perturbation is due to *structural* distortion, and not simply due to differences in surface form words. For example, given the original sentence "I ate an apple" and the perturbed sentence "I an ate apple", without the permutation matrix we would be subtracting the (contextualized) representation for "an" from the representation for "ate", which results it greater distortion because these are different words. *Z-scoring.* The z-scoring is essential for ensuring that increases or decreases in distortion are not due just to uniform scaling. Consider a pathological case where the representation in each layer of BERT is just a scaled and rotated version of the embedding layer (Fig. 6R3; https://i.imgur.com/K8ZATmH.png). Naïve computation of the distances is fooled by this, but z-scoring is not (left figure). This is important because, empirically, the variance of activity in BERT layers is not constant, and without z-scoring the resulting distances pick up on this uninteresting variance (middle figure). We also note that z-scoring is not the only way to make the distortion metric scale-invariant: we can also use the average vector norms or the total variance (right figure).
>
> - **“However, this is also a complementary weakness in that the paper doesn't add anything particularly surprising given prior work in this area.”** While there is indeed some evidence that BERT-like models are able to encode hierarchical syntactic information in its hidden layers (Liu et al. 2019; Tenney et al. 2019; Hewitt and Manning 2019), the field has yet to reach a consensus on (1) methodologies by which to extract such information (prior work has mostly relied on supervised probes which is known to be problematic (Hewitt and Liang 2019)) and (2) the mechanisms by which such information is encoded. We believe that our approach, which measures the change in distributed representations following targeted interventions to specific input structures, is novel, and quite different from the existing work on supervised probes. Finally, even if the phenomenological results are mainly in line findings from existing work on supervised probes, we believe that arriving at these results via a new method is interesting/important!
>
> - **“I would have also liked to see some treatment of other pretrained models…”** Thanks for the suggestion. Please see Figure 7 and 8 of the supplementary materials, where we provide experiments run on other pretrained models such as ALBERT, RoBERTa, DistilBERT and XLNet. We observe largely similar trends across the different models. We chose to focus our attention on pretrained bidirectional transformer models (i.e. BERT and its variants) given their popularity. Performing similar types of analysis on unidirectional language models such as GPT2 is challenging because it is not clear that the hidden representation $h_t$ at time $t$ is the "right" representation for $x_t$ (i.e. $t$-th word in the sentence). This is because unlike bidirectional models where $h_t$ is used to both encode *and* decode $x_t$, in unidirectional models $h_t$ is used to encode $x_t$ but not decode $x_t$. We think this is an interesting future direction however, and am currently exploring ways to export our perturbation analysis to unidirectional models.
>
> - **“Alternatively, the paper could have take the approach of developing the best overall probe to detect what is captured by a model”** This is an important open issue in the field. As the reviewer notes, while there are different types of probes that have been proposed to analysis BERT-style models, there hasn't been enough work done to compare and contrast different methods, and give accountings of when certain probes should (or should not) be used. While important however, we think that this is orthogonal to our main contribution, which introduces a novel approach to detecting representational correlates of linguistic phenomena in neural models though controlled input perturbations.

---

### Author Response · Authors · 2020-11-24
**Overall response**

We thank all the reviewers for their comments, and also suggesting many insightful experiments. We have updated the paper to include the additional experiments (page 9), and we believe they have significantly improved the paper. We have also responded to each reviewer's individual questions and have provided imgur links to the individual figures (as well as explanations for these figures) in our reviewer-specific rebuttals.

To summarize, our additional experiments include:

- (R2) Rank correlation between distortion and tree distance for untrained model as a baseline to compare against
- (R2) Distortion experiments when subword tokens are not averaged
- (R2) Analysis of change in attention for swapped words
- (R2) Baseline likelihoods for our supervised probes
- (R3) Simulations on scaling the embedding layers to show importance of z-score normalization
- (R4) Distortion analysis by NP vs VP swaps, including multivariate analysis to control for swap size
- (R5) Distortion analysis by n-gram chunk size (against taking into account swap size) to show that n-gram swap trend cannot be explained by simple positional differences.


Further, several reviewers have noted that many of the findings are not surprising because there have been many papers which show that BERT is encoding syntax in its hidden layers. Indeed, many papers do show this (Liu et al. 2019; Tenney et al. 2019). However, much existing work has focused on supervised probes, and we believe that our approach, which measures the change in distributed representations following targeted interventions to specific input structures, is novel, and quite different from the existing work. **Our approach further enables interesting additional analyses that may not be readily possible with supervised probes (e.g. more fine-grained analysis via conditioning on tree distance)**. Finally, even if the phenomenological results are mainly in line findings from existing work on supervised probes, we believe that arriving at these results via a new method is interesting/important.

---

### Decision · Program_Chairs · 2021-01-07
**Final Decision**

**Decision:**

Reject

**Comment:**

The paper presents a significant body of seemingly solid work, but its contribution nevertheless feels limited: It evaluates a single MLM on a single dataset, and results are largely unsurprising. Note: The authors added experiments on other LMs in the rebuttal. The idea of using perturbations is related in spirit to many interpretability methods and adversarial techniques, and using higher-order correlations for interpreting neural networks is, for example, at the heart of relational similarity analysis. A few suggestions to make the work more relevant to a wider audience: Compare with several probing techniques - e.g., in a tree-decoding set-up - or contrast results across domains (using OntoNotes), or across languages (using OntoNotes and other PTB-style treebanks). Also: While results were added for multiple LMs, differences were not analysed in detail.